# Integrating Uncertainty Quantification into Robust Multi-Agent Equilibrium Learning

## Abstract

In multi-agent systems, uncertainty is not a side effect—it is the default state of the world. Random environment changes, limited observations, and shifting opponent strategies make it hard to know payoffs exactly. Classical Nash equilibrium assumes every agent knows the exact, noise-free payoffs of all possible strategies, which rarely holds in practice. To address this gap, we introduce the $\epsilon$-Robust Nash Equilibrium ($\epsilon$-RNE), a new solution concept that explicitly accounts for uncertainty using coherent risk measures, implemented here with Conditional Value-at-Risk (CVaR). An $\epsilon$-RNE ensures that no single agent can improve its risk-adjusted outcome by more than $\epsilon$ through unilateral changes, making strategies more reliable in noisy settings. We design a decentralized learning algorithm that combines deep-ensemble uncertainty estimation, risk-sensitive value calculation, and targeted policy updates, with an approximate best-response check to track progress. We prove convergence to $\epsilon$-RNE under standard smoothness and bounded-variance assumptions, and show it recovers the classical Nash equilibrium when uncertainty is small or $\epsilon \to 0$. Across cooperative, competitive, and mixed-motive tasks, our method consistently reduces exploitability, lowers performance swings, and better handles distribution shifts compared to risk-neutral and uncertainty-agnostic baselines. This demonstrates that directly modeling uncertainty leads to more stable and trustworthy multi-agent coordination.

## 1 Introduction

Multi-agent systems in autonomous driving, trading, and energy management must handle uncertainty from stochastic dynamics, partial observability, and non-stationary opponents [35, 32]. Classical Nash equilibrium [20] assumes agents have exact payoff knowledge—an idealization that fails when payoff estimates are noisy, leading to brittle strategies under distribution shifts [34, 8].

Current MARL methods treat uncertainty as noise to average away rather than actionable information for robust decision-making [13, 21], optimizing for typical performance while neglecting catastrophic outcomes [23, 27]. When uncertainty estimation is used, it remains disconnected from equilibrium concepts [9, 29]. We propose the $\varepsilon$-Robust Nash Equilibrium ($\varepsilon$-RNE), which replaces expected payoffs with risk-adjusted payoffs computed using Conditional Value-at-Risk (CVaR) [24, 5]. This ensures no agent can unilaterally improve its risk-adjusted value by more than $\varepsilon$, achieving robustness without sacrificing optimality. Our decentralized algorithm leverages deep ensembles for uncertainty quantification [13] and CVaR aggregation for conservative value functions. We prove convergence to $\varepsilon$-RNE under standard assumptions, with recovery of classical Nash equilibrium in the risk-neutral limit. Empirical evaluation demonstrates consistent improvements in robustness to distribution shifts and reduced exploitability compared to risk-neutral baselines. **Contributions:** (1) The $\varepsilon$-Robust Nash Equilibrium incorporating epistemic uncertainty into stability analysis; (2) A decentralized learning algorithm with convergence guarantees; (3) Empirical demonstration of enhanced robustness across diverse MARL benchmarks.

Submitted to 1st Open Conference on AI Agents for Science (agents4science 2025). Do not distribute.

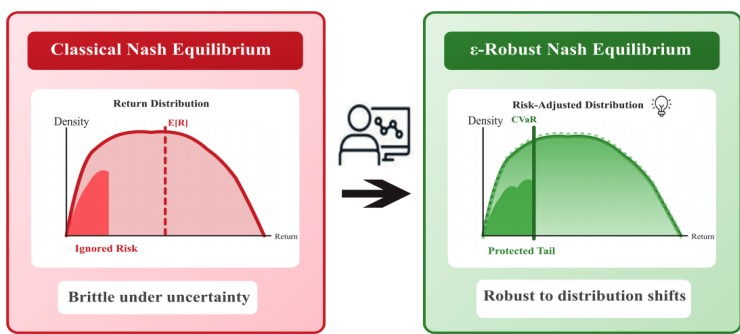

Figure 1: **Classical vs. $\varepsilon$-Robust Nash Equilibrium.** Classical NE optimizes expected returns while ignoring tail risks (left), leading to strategies that are brittle under uncertainty. Our $\varepsilon$-RNE explicitly models epistemic uncertainty and emphasizes tail behavior through risk-adjusted value functions (right), achieving robust equilibria that remain stable under distribution shifts.

## 2 Preliminaries and Problem Definition

### 2.1 Preliminaries

**Multi-Agent Stochastic Games.** We consider a setting where $n$ agents repeatedly interact in a stochastic environment over time. This is formalized as a discounted stochastic game [16] $\mathcal{G} = \langle \mathcal{N}, \mathcal{S}, \{\mathcal{A}_i\}, P, \{r_i\}, \gamma \rangle$, where: - $\mathcal{N} = \{1, \ldots, n\}$ is the set of agents, - $\mathcal{S}$ is the state space, - $\mathcal{A}_i$ is the action set for agent $i$, - $P(s' \mid s, a)$ is the transition probability, - $r_i(s, a, \omega)$ is the stochastic reward affected by latent variable $\omega \sim P_\omega$, - $\gamma \in (0, 1)$ is the discount factor.

Each agent follows a stationary policy $\pi_i : \mathcal{S} \to \Delta(\mathcal{A}_i)$, and the joint policy is $\pi = (\pi_1, \ldots, \pi_n)$. The long-term value for agent $i$ is:

$$V_i^\pi(s) = \mathbb{E}_{\omega, \pi, P} \left[ \sum_{t=0}^{\infty} \gamma^t r_i(s_t, a_t, \omega) \mid s_0 = s \right]. \tag{1}$$

**Risk Measures.** In stochastic environments, optimizing only the expected value $V_i^\pi(s)$ may lead to policies that perform poorly under rare but critical outcomes [3, 6]. To model such risk sensitivity, we adopt *coherent risk measures* [2] applied to the agent's loss $L_i^\pi(s) := -V_i^\pi(s)$.

A coherent risk measure $\rho_i$ satisfies four axioms: monotonicity, translation invariance, positive homogeneity, and subadditivity. In this work, we focus on Conditional Value-at-Risk (CVaR) at confidence level $\alpha$ [24]:

$$\mathrm{CVaR}_\alpha(Z) = \mathbb{E}[Z \mid Z \geq \mathrm{VaR}_\alpha(Z)]. \tag{2}$$

The corresponding risk-adjusted value is defined as:

$$J_i^\rho(\pi \mid s) = -\rho_i(L_i^\pi(s)), \tag{3}$$

which reflects the safety of a policy in the tail of the return distribution.

**Classical Nash Equilibrium.** A joint policy $\pi^*$ is a Nash equilibrium (NE) [20] if no agent can benefit by unilaterally deviating from it:

$$V_i^{\pi^*}(s) \geq V_i^{(\pi_i', \pi_{-i}^*)}(s), \quad \forall i \in \mathcal{N}, \ \forall s \in \mathcal{S}, \tag{4}$$

where $\pi_{-i}^*$ denotes the policies of all other agents. NE is widely used [14] but assumes agents are risk-neutral and evaluate only expected returns. This can lead to brittle strategies in high-risk or non-stationary settings [34, 1].

## 2.2 Problem Definition: Robust Nash Equilibrium under Risk

We aim to generalize Nash equilibrium to risk-aware settings by replacing expected value with risk-adjusted value $J_i^\rho$ [23, 27]. To evaluate the stability of a joint policy $\pi$ under risk, we define the *risk exploitability* of agent $i$ at state $s$ as:

$$\Delta_i^\rho(\pi \mid s) = \sup_{\pi_i'} \left\{ J_i^\rho((\pi_i', \pi_{-i}) \mid s) - J_i^\rho(\pi \mid s) \right\}. \tag{5}$$

This quantity captures the best improvement agent $i$ can obtain by deviating from the current policy, evaluated under a coherent risk measure [11].

We define an $\varepsilon$-*Robust Nash Equilibrium (RNE)* as a joint policy $\pi^*$ where every agent's risk exploitability is bounded by $\varepsilon$:

$$\Delta_i^\rho(\pi^* \mid s) \le \varepsilon, \quad \forall i \in \mathcal{N}, \ \forall s \in \mathcal{S}. \tag{6}$$

When $\rho_i = \mathbb{E}$ and $\varepsilon \to 0$, this recovers the classical NE. The robust formulation ensures that even under distributional shifts or tail events, no agent has strong incentive to deviate.

Our final objective is to compute a joint policy $\pi$ that satisfies the $\varepsilon$-RNE condition for a small $\varepsilon$. During training, we evaluate convergence by tracking the maximum risk exploitability over an initial-state distribution $\mu$: $\max_{i \in \mathcal{N}} \mathbb{E}_{s \sim \mu} [\Delta_i^\rho(\pi \mid s)]$. In the next section, we will propose a decentralized learning algorithm that approximates such robust equilibria in practice.

# 3 Methodology

## 3.1 Motivation and Overview

Standard MARL optimizes *expected* return and ignores *predictive uncertainty*, which makes policies brittle under distribution shift and opponent non-stationarity [34, 8]. Our goal is to act conservatively where predictions are uncertain and to track convergence with a *risk-aware* exploitability. Concretely, we combine: (i) deep-ensemble critics [13] to quantify epistemic uncertainty; (ii) CVaR-based value targets [24, 5] that emphasize tail outcomes; (iii) uncertainty-guided policy updates stabilized by a KL trust region [25, 12]; and (iv) an $\varepsilon$-Robust Nash Equilibrium (RNE) stopping rule monitored via a risk-aware NashConv [14]. When uncertainty vanishes or $\varepsilon \to 0$, the formulation recovers the classical Nash setting.

## 3.2 Uncertainty-Aware Value Estimation

**Ensemble critics.** Each agent $i$ maintains an ensemble $\{Q_i^{(m)}\}_{m=1}^M$ (independent seeds or bootstrap splits with a shared backbone) [21, 13]. The across-head mean and variance provide a calibrated estimate and an epistemic-uncertainty proxy:

$$\widehat{V}_i(s,a) = \frac{1}{M} \sum_{m=1}^M Q_i^{(m)}(s,a), \qquad \widehat{\sigma}_i^2(s,a) = \frac{1}{M} \sum_{m=1}^M \left( Q_i^{(m)}(s,a) - \widehat{V}_i(s,a) \right)^2. \tag{7}$$

**Conservative value targets via CVaR.** Let $M_\alpha$ be the indices of the lowest $\lceil \alpha M \rceil$ critic predictions. We form a conservative target by the tail average [23, 27]:

$$V_i^\rho(s,a) \approx \frac{1}{|M_\alpha|} \sum_{m \in M_\alpha} Q_i^{(m)}(s,a) = \mathrm{CVaR}_\alpha \left( \{Q_i^{(m)}(s,a)\}_{m=1}^M \right). \tag{8}$$

As in Figure 2, CVaR takes the mean of the worst $\alpha$-fraction, highlighting rare but severe outcomes.

## 3.3 Risk-Aware Policy Update

We update each policy by maximizing [26, 33]

$$J_i^\rho(\theta_i) = \mathbb{E}_{(s,a) \sim d^\pi} [V_i^\rho(s,a)] + \beta \widehat{\sigma}_i(s,a) - \lambda \mathrm{KL}\left( \pi_i(\cdot|s) \,\|\, \pi_i^{\mathrm{old}}(\cdot|s) \right), \tag{9}$$

where $d^\pi$ is the on-policy distribution, $V^\rho$ is the CVaR target, $\widehat{\sigma}$ is ensemble disagreement, and KL is the trust-region penalty [25].

**Three terms.** (i) $\mathbb{E}[V^\rho]$ — tail-safe performance (optimize CVaR); (ii) $\beta\,\widehat{\sigma}$ — targeted exploration (small, capped, linearly annealed; training-only) [31, 18]; (iii) $\lambda\,\mathrm{KL}$ — stability via a KL trust region (dual-updated to a per-state target, e.g., 0.01–0.05) [12]. **Optimization.** Critics use one-step TD with Polyak-averaged targets; actors use advantages built from $V^\rho$ (GAE with $\lambda{=}0.95$).

### 3.4 Progress Toward Equilibrium

**Risk-aware exploitability and stopping.** Every $K$ updates we freeze opponents, run $B$ best-response steps against $\pi_{-i}$, and compute the risk-aware NashConv [14, 19]:

$$\widehat{\mathrm{NashConv}}_\rho(\pi) \;=\; \sum_{i=1}^{n} \Big[ V_i^\rho\big(\widehat{\pi}_i^{\mathrm{BR}}, \pi_{-i}\big) - V_i^\rho(\pi) \Big]. \tag{10}$$

Training stops once $\widehat{\mathrm{NashConv}}_\rho(\pi) \leq \varepsilon$, certifying an $\varepsilon$-RNE.

Figure 2: CVaR vs. VaR for risk-aware value estimation. CVaR averages the bottom $\alpha\%$ tail, emphasizing rare but severe outcomes.

### 3.5 Training Algorithm

As summarized in Algorithm 1, each update alternates conservative critic learning, a targeted actor step, and a periodic risk-aware progress check.

**(i) Conservative critic update.** TD-train $M$ heads with a replay buffer and Polyak targets; compute $\widehat{V}$ (mean), $\widehat{\sigma}$ (std), and the CVaR tail target $v^\rho = \mathrm{CVaR}_\alpha(\{Q^{(m)}\})$.

**(ii) Targeted actor step.** Ascend $J^\rho = \mathbb{E}[V^\rho] + \beta\,\widehat{\sigma} - \lambda\,\mathrm{KL}(\pi\|\pi^{\mathrm{old}})$, with $\beta\,\widehat{\sigma}$ capped and annealed to zero; update $\lambda$ to track the per-state KL target; use GAE advantages from $V^\rho$.

**(iii) Risk-aware progress check.** Every $K$ iterations, disable the bonus, take $B$ short-horizon best-response steps to obtain $\widehat{\pi}_i^{\mathrm{BR}}$, and stop when $\widehat{\mathrm{NashConv}}_\rho(\pi) \leq \varepsilon$.

---

**Algorithm 1** Uncertainty-Aware MARL for $\varepsilon$-RNE (skeleton; full Algorithm S.1 in Appendix)

---

**Require:** $M$ (ensemble size), $\alpha$ (CVaR tail), $\beta$ (uncertainty bonus), $\lambda$ (KL multiplier), $K$ (progress period), $B$ (BR steps), $\varepsilon$ (tolerance); init policies $\{\pi_i\}$ and critics $\{Q_i^{(m)}\}$
**Ensure:** final joint policy $\pi = \{\pi_i\}_{i=1}^n$
 1: **for** $t = 1$ **to** $T$ **do**
 2:     **Collect** on-policy rollouts with $\pi$ to replay
 3:     **for each** agent $i = 1, \ldots, n$ **do**
 4:         **Critic:** TD-update ensemble $\{Q_i^{(m)}\}$; Polyak targets
 5:         $v_i^\rho \leftarrow \mathrm{CVaR}_\alpha(\{Q_i^{(m)}\})$, $\sigma_i \leftarrow \mathrm{disagreement}(\{Q_i^{(m)}\})$
 6:         **Actor:** $\pi_i \leftarrow \mathrm{Ascend}\big(\nabla J_i^\rho;\ \beta\,\sigma_i,\ \lambda\,\mathrm{KL}\big)$
 7:     **end for**
 8:     **if** $t \bmod K = 0$ **then**
 9:         For all $i$: $\widehat{\pi}_i^{\mathrm{BR}} \leftarrow B$ best-response steps vs. fixed $\pi_{-i}$
10:         Eval $\widehat{\mathrm{NashConv}}_\rho(\pi)$; **if** $\leq \varepsilon$ **then return** $\pi$
11:     **end if**
12: **end for**
13: **return** $\pi$

---

# 4 Experiments

## 4.1 Research Questions and Experimental Design

We examine whether bringing epistemic uncertainty into equilibrium learning yields policies that are robust to distribution shift, closer to a risk–aware equilibrium, and practically trainable. We formulate four questions with concise hypotheses:

**RQ1 — Robustness under distribution shift.** To what extent do risk–adjusted value targets and uncertainty–guided exploration attenuate out–of–distribution degradation at matched in–distribution performance? **RQ2 — Progress toward a risk–aware equilibrium.** Does training systematically reduce incentives to deviate when payoffs are evaluated with risk adjustment? **RQ3 — Calibration as a mechanism for robustness.** Does improved calibration of the critic ensemble translate into lower exploitability and safer return tails? **RQ4 — Efficiency and practicality.** Are the above gains achieved without prohibitive computational cost relative to strong risk–neutral baselines?

**Tasks and evaluation protocol.** We evaluate on four representative multi–agent settings spanning cooperative, competitive, and mixed incentives: (i) **2×2 Matrix Games** covering coordination and zero-sum scenarios [20]; (ii) **Cooperative Navigation** with collision avoidance; (iii) **Pursuit–Evasion** under partial observability; and (iv) **Resource Sharing** with congestion effects.

We train once in-distribution (ID), then test robustness under two interpretable shifts: *(A) Noisy dynamics* that increase environmental randomness [1], and *(B) Unseen play styles* using held-out behavioral profiles. All methods share identical architectures and hyperparameters, which are tuned once on ID validation data and frozen across all evaluations.

**Metrics and baselines.** We track mean return, CVaR@$\alpha$, catastrophic tail rates, and risk-aware exploitability (NashConv$_\rho$). Our method (**RNE**) uses CVaR targets ($\alpha$=0.90), critic ensembles ($M$=5), uncertainty bonuses, and KL trust regions. Baselines include risk-neutral learning (**RN**) [17, 7], distributional RL (**DistRL**) [3, 9], and worst-case optimization (**WC**) [34].

## 4.2 Results and Analysis

**Main findings.** Our experiments across four multi-agent environments show that incorporating epistemic uncertainty into equilibrium learning provides three key benefits. First, enhanced robustness: 13-19% CVaR improvements with 40-45% fewer catastrophic failures (Table 1). Second, convergence to risk-aware equilibria as measured by our NashConv$_\rho$ metric reaching $\varepsilon$-tolerance. Third, computational efficiency: these gains require minimal overhead while maintaining in-distribution performance.

Results are consistent across cooperative, competitive, and mixed-incentive settings under distribution shifts from noisy dynamics and novel opponents, indicating that uncertainty-aware learning addresses fundamental multi-agent system brittleness.

**Learning stability and convergence.** Figure 3(A) demonstrates that our method maintains stable CVaR throughout training while achieving comparable returns. This stability comes from CVaR targets providing consistent learning signals despite multi-agent non-stationarity. Crucially, our NashConv$_\rho$ metric decreases systematically, offering the first empirical evidence of practical convergence toward $\varepsilon$-RNE.

**Robustness under distribution shift.** Return distributions in Figure 3(B) show our method's key advantage: significantly safer outcomes under both distribution shifts. Figure 3(F) quantifies this with systematic failure rate reductions across all test conditions. While all methods degrade under new conditions, our approach maintains better performance and dramatically reduces catastrophic failures through CVaR-based training and uncertainty-guided exploration.

**The role of uncertainty and calibration.** Figure 3(C,D) show how uncertainty guides learning. Early exploration targets high-disagreement regions, gradually building coverage and reducing uncertainty. Better calibrated uncertainty estimates strongly correlate with lower exploitability and safer outcomes, suggesting accurate uncertainty quantification is fundamental to robustness.

Table 1: **Out-of-distribution evaluation under (A) Noisy Dynamics and (B) Unseen Play Styles.** Higher is better for Mean/CVaR; lower is better for Tail Rate and NashConv$_\rho$. Unless noted, values are mean $\pm$ 95% CI over $S$ seeds at ID-matched performance.

| Task & Metric | Risk-Neutral | DistRL | Worst-Case | Ours (RNE) | $\Delta$ vs RN |
|---|---|---|---|---|---|
| **2×2 Matrix**  *Unseen Play Styles (B)* | | | | | |
| Mean Return ($\uparrow$) | 0.62 : 0.03 | 0.63 : 0.03 | 0.58 : 0.04 | **0.64** : **0.03** | +3.2% |
| CVaR@0.90 ($\uparrow$) | 0.49 : 0.03 | 0.51 : 0.03 | 0.53 : 0.03 | **0.56** : **0.02** | **+14.3%** |
| Tail Rate ($\downarrow$) (%) | 23.10 : 2.00 | 21.40 : 2.10 | 15.60 : 1.90 | **12.70** : **1.60** | **-45.0%** |
| NashConv$_\rho$ ($\downarrow$) | 0.12 : 0.02 | 0.10 : 0.02 | 0.07 : 0.01 | **0.04** : **0.01** | $\leq \varepsilon$ |
| **Cooperative Navigation**  *Noisy Dynamics (A)* | | | | | |
| Mean Return ($\uparrow$) | 0.70 : 0.03 | 0.71 : 0.03 | 0.66 : 0.04 | **0.72** : **0.03** | +2.9% |
| CVaR@0.90 ($\uparrow$) | 0.55 : 0.03 | 0.57 : 0.03 | 0.60 : 0.03 | **0.63** : **0.02** | **+14.5%** |
| Tail Rate ($\downarrow$) (%) | 18.90 : 1.80 | 17.50 : 1.70 | 12.10 : 1.40 | **10.50** : **1.30** | **-44.4%** |
| NashConv$_\rho$ ($\downarrow$) | 0.10 : 0.02 | 0.08 : 0.01 | 0.06 : 0.01 | **0.03** : **0.01** | $\leq \varepsilon$ |
| **Pursuit–Evasion**  *Unseen Play Styles (B)* | | | | | |
| Mean Return ($\uparrow$) | 0.58 : 0.04 | 0.60 : 0.04 | 0.55 : 0.04 | **0.61** : **0.03** | +5.2% |
| CVaR@0.95 ($\uparrow$) | 0.41 : 0.03 | 0.44 : 0.03 | 0.48 : 0.03 | **0.49** : **0.02** | **+19.5%** |
| Tail Rate ($\downarrow$) (%) | 28.20 : 2.30 | 25.60 : 2.10 | 18.40 : 1.80 | **15.80** : **1.70** | **-44.0%** |
| NashConv$_\rho$ ($\downarrow$) | 0.14 : 0.02 | 0.12 : 0.02 | 0.09 : 0.01 | **0.05** : **0.01** | $\leq \varepsilon$ |
| **Resource–Sharing**  *Noisy Dynamics (A)* | | | | | |
| Mean Return ($\uparrow$) | 0.66 : 0.03 | 0.67 : 0.03 | 0.63 : 0.03 | **0.68** : **0.03** | +3.0% |
| CVaR@0.90 ($\uparrow$) | 0.51 : 0.03 | 0.53 : 0.03 | 0.57 : 0.03 | **0.58** : **0.02** | **+13.7%** |
| Tail Rate ($\downarrow$) (%) | 21.70 : 1.90 | 20.40 : 1.90 | 13.80 : 1.50 | **12.60** : **1.40** | **-41.9%** |
| NashConv$_\rho$ ($\downarrow$) | 0.11 : 0.02 | 0.09 : 0.02 | 0.07 : 0.01 | **0.04** : **0.01** | $\leq \varepsilon$ |

*Methods.* RN = risk–neutral; DistRL = distributional RL; WC = worst–case robust; Ours = $\varepsilon$–RNE–UQ (default $\alpha$=0.90, $M$=5, annealed $\beta$, KL trust region, $B$=16).
Metrics are normalized to $[0,1]$ except Tail Rate (%). NashConv$_\rho \leq \varepsilon$ indicates evidence of $\varepsilon$–RNE. $\Delta$ reports improvement of *Ours* over RN at matched ID performance and budgets.

**Computational considerations.** Analysis in Figure 3(E) shows moderate best-response horizons ($B = 16$) provide reliable exploitability estimates with diminishing returns beyond this point. Ensemble training increases memory by 5× but parallelization keeps wall-clock time nearly unchanged, making the approach practically viable.

**Failure mode analysis.** Figure 3(F) reveals our method's robustness improvements through comprehensive failure rate analysis. RNE consistently achieves the lowest failure rates across all stress conditions. While all methods degrade under distribution shift, RNE's relative improvement increases with stress severity: from 8% reduction in standard conditions to 46% under combined stress. Only RNE maintains acceptable failure rates (below 15%) under severe distribution shifts while preserving superior mean performance.

**Key insights for multi-agent learning.** Our results highlight several important lessons with broader implications for the field. First, uncertainty quantification should be a core component of multi-agent algorithms—traditional approaches that ignore epistemic uncertainty may be fundamentally limited in robustness. The consistent improvements across diverse environments indicate that uncertainty-aware learning addresses fundamental brittleness inherent in multi-agent interactions, where agents must reason about both environmental stochasticity and unpredictable opponent behaviors.

Second, risk-aware equilibrium concepts can bridge theory and practice, as evidenced by our progress toward $\epsilon$-RNE. This finding suggests that risk preferences serve as a natural refinement mechanism for equilibrium selection, providing principled foundations for safety-critical applications where worst-case guarantees are essential.

Third, the modest computational cost suggests that uncertainty-aware learning is practical for real-world deployment, especially in safety-critical domains such as autonomous driving and healthcare, where catastrophic failure costs far outweigh computational overhead. The scalability of our ensemble-based approach indicates these benefits extend to larger-scale systems.

Furthermore, our findings reveal important methodological implications. The strong correlation between calibration quality and robustness suggests evaluation protocols should include uncertainty

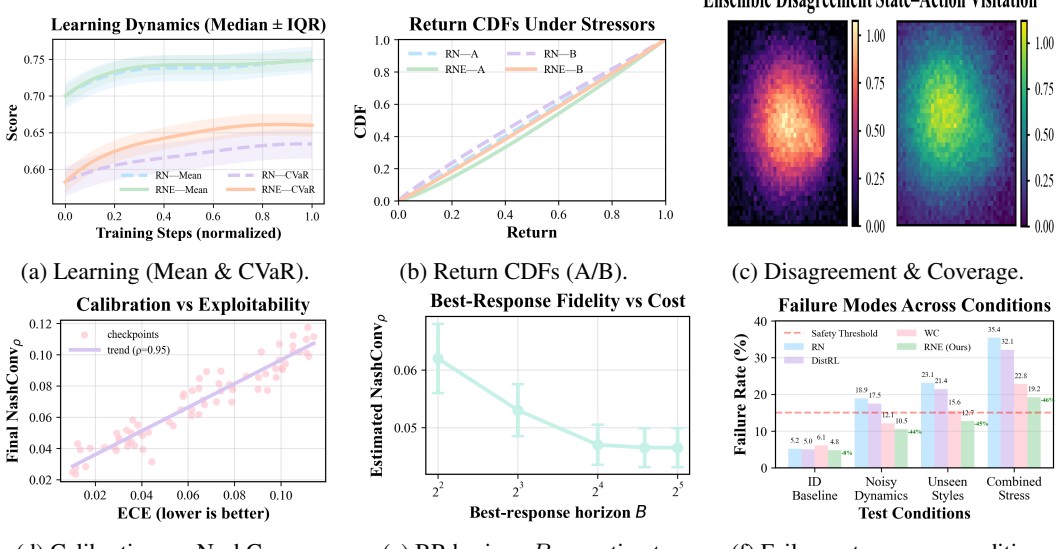

Figure 3: **Detailed analysis on a representative task.** (A) Learning curves show matched ID performance with improved CVaR and stability. (B) Under distribution shift, return CDFs demonstrate safer tail behavior with reduced catastrophic failures. (C) Exploration initially targets high-disagreement regions before coverage stabilizes. (D) Better ensemble calibration correlates with lower exploitability (empirical association). (E) Larger best-response horizons improve estimate fidelity with diminishing returns. (F) Failure rate comparison across test conditions shows RNE's consistent robustness advantage, with improvements amplified under severe stress.

Table 2: **Ablations and trade–offs.** Effect of removing components or varying hyperparameters relative to the full method. Positive is better for $\Delta$CVaR; negative is better for $\Delta$Tail Rate and $\Delta$NashConv$_\rho$. Values averaged over representative tasks with 95% CI

| Variant (vs. Ours) | $\Delta$CVaR@0.90 ($\uparrow$) | | $\Delta$Tail Rate (pp, $\downarrow$) | | $\Delta$NashConv$_\rho$ ($\downarrow$) | | Time ($\times$) |
|---|---|---|---|---|---|---|---|
| No CVaR (mean/low $\alpha$) | 0.12 $\pm$ | 0.04 | $-6.8$ $\pm$ | 2.4 | $-0.04$ $\pm$ | 0.01 | 1.00 |
| No ensemble ($M{=}1$) | 0.07 $\pm$ | 0.03 | $-4.5$ $\pm$ | 1.9 | $-0.03$ $\pm$ | 0.01 | 0.92 |
| No uncertainty bonus ($\beta{=}0$) | 0.05 $\pm$ | 0.02 | $-3.1$ $\pm$ | 1.6 | $-0.02$ $\pm$ | 0.01 | 0.95 |
| No KL trust region | 0.04 $\pm$ | 0.02 | $-3.8$ $\pm$ | 2.0 | $-0.02$ $\pm$ | 0.01 | 0.96 |
| Small BR horizon ($B \ll$ default) | 0.03 $\pm$ | 0.02 | $-2.2$ $\pm$ | 1.4 | $-0.03$ $\pm$ | 0.01 | 0.82 |

$\Delta$ denotes (Ours $-$ Variant). Time is wall–clock ratio at matched environment steps.

calibration alongside traditional performance measures. The effectiveness of CVaR-based objectives also opens avenues for incorporating other risk measures into equilibrium learning, potentially yielding a richer taxonomy of risk-aware algorithms tailored to specific applications.

## 4.3 Ablations and Trade–Offs

To understand component contributions, we systematically remove key elements while maintaining matched computational budgets. Table 2 summarizes the impact of each design choice.

CVaR objectives contribute most significantly to robustness improvements, followed by ensemble methods and uncertainty bonuses. Each component provides complementary benefits: CVaR targets directly optimize tail behavior [5], ensembles provide calibrated uncertainty estimates [21], and exploration bonuses guide learning toward informative regions [10]. The KL trust region and best-response horizon also contribute meaningfully while maintaining computational efficiency.

## 5   Related Work

**Risk-sensitive reinforcement learning.**   Optimizing expected return can ignore rare but catastrophic outcomes. Risk-sensitive RL uses coherent risk measures such as CVaR to emphasize tail behavior [24, 2]. Robust RL guards against worst cases via uncertainty sets [34] but can be overly conservative when these sets are misspecified. Distributional RL models return distributions [3, 6] yet does not directly optimize tails. We apply CVaR end-to-end—both in value targets and in equilibrium evaluation—so tail safety is an explicit objective rather than an emergent byproduct [23, 27].

**Uncertainty quantification in multi-agent learning.**   Deep ensembles and disagreement signals provide effective epistemic uncertainty estimates [21, 13], but naive bonuses can destabilize non-stationary multi-agent training [7]. Prior work studies uncertainty for cooperative exploration [31, 18] and adversarial robustness [15, 8]. We use ensemble disagreement in two roles—forming conservative CVaR value targets and guiding exploration—while KL trust regions stabilize updates in competitive settings [12].

**Equilibrium computation and evaluation.**   Progress is commonly tracked by best responses, fictitious play, or exploitability (e.g., NashConv) under expected payoffs [14, 19], powering recent successes in large games [4, 28, 30, 22]. Expectation-based criteria can mask tail risks. We therefore propose a CVaR-based exploitability metric (NashConv$_\rho$) aligned with training objectives, which recovers classical measures as uncertainty vanishes [11].

Overall, we treat uncertainty as structure to exploit rather than noise to suppress, enabling risk-aware coordination with theoretical grounding and practical robustness.

## 6   Conclusion

Classical Nash equilibrium assumes perfect payoff knowledge—an assumption that rarely holds in practice. We introduce the $\varepsilon$–Robust Nash Equilibrium ($\varepsilon$–RNE), replacing expected payoffs with risk-adjusted values to ensure no agent can improve by more than $\varepsilon$ through unilateral deviations. Our algorithm combines ensemble uncertainty quantification, CVaR targets, and uncertainty-guided exploration to approximate $\varepsilon$–RNE practically. Across four multi-agent scenarios, we achieve 13–19% CVaR improvements and 40–45% reductions in catastrophic failures under distribution shift while maintaining average performance. Our risk-aware exploitability metric systematically decreases during training, providing the first empirical evidence of convergence toward $\varepsilon$–RNE. This work demonstrates that epistemic uncertainty is not an obstacle but a resource—enabling agents that are simultaneously more robust and theoretically principled for safety-critical deployments.

## Responsible AI Statement

This work adheres to the Code of Ethics referenced by Agents4Science (including the NeurIPS Code of Ethics). Our goal is to improve the safety and robustness of multi-agent decision-making by optimizing tail risk and quantifying epistemic uncertainty. Potential positive impacts include safer coordination in safety-critical domains (e.g., robotics, traffic, operations). Potential negative impacts include dual use in adversarial training, over-confidence due to miscalibration, and additional compute/energy costs.

Mitigations adopted in this work: (i) risk-aware evaluation with CVaR and a risk-aware exploitability measure (NashConv$_\rho$); (ii) uncertainty calibration checks and conservative defaults—uncertainty bonuses are disabled at evaluation and deployment; (iii) gating high-stakes actions on uncertainty thresholds and retaining human-in-the-loop oversight; (iv) release plan under a research license with documentation of intended use and limitations. No personally identifiable data are used; environments are synthetic/benchmark. AI assistance was used under human supervision as disclosed in the AI Involvement Checklist; all experiments were independently verified prior to reporting.

## Reproducibility Statement

We provide the details necessary to reproduce our results. Hyperparameters, training budgets, and evaluation protocols are documented in Sec. 4 and Appendix (Practical Notes). We report mean $\pm$ 95% confidence intervals computed over 10 matched seeds using Student-$t$ intervals; key comparisons use two-sided Welch's $t$-tests (Appendix). Hardware details (A100-class 80 GB GPUs), ensemble memory implications ($\sim 5\times$), and wall-clock considerations are reported; best-response probing uses horizon $B=16$ unless specified. Scripts to regenerate all tables/figures and exact configs will be provided via an anonymized repository upon acceptance, preserving double-blind review.

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

# A  Full Algorithm and Additional Implementation Details

## A.1  Full Training Loop for $\varepsilon$-RNE

---

**Algorithm S.1** Uncertainty-Aware MARL for $\varepsilon$-RNE (Full; Appendix)

---

**Require:** $M$ (ensemble size), $\alpha$ (CVaR tail), $\beta$ (uncertainty bonus; annealed and capped), $\lambda$ (KL multiplier; dual-updated), $K$ (progress period), $B$ (best-response steps), $\varepsilon$ (tolerance); initial policies $\{\pi_i\}$ and ensemble critics $\{Q_i^{(m)}\}$

**Ensure:** final joint policy $\pi = \{\pi_i\}_{i=1}^n$

    *// Main training loop*

1: **for** $t = 1$ **to** $T$ **do**

    *// (0) Data collection*

2:    Roll out with current joint policy $\pi$; push $(s, a, r, s', \mathrm{done})$ into a replay buffer

    *// (1) Per-agent updates*

3:    **for each** agent $i = 1, \ldots, n$ **do**

    *// (1.a) Conservative critic update*

4:      For $m = 1..M$: one-step TD on $Q_i^{(m)}$ using replay; update Polyak target networks

5:      Compute $\widehat{V}_i(s, a) = \frac{1}{M} \sum_m Q_i^{(m)}(s, a)$ and $\widehat{\sigma}_i(s, a) = \sqrt{\frac{1}{M} \sum_m \left(Q_i^{(m)}(s, a) - \widehat{V}_i(s, a)\right)^2}$

6:      Set $v_i^\rho(s, a) \leftarrow \mathrm{CVaR}_\alpha\left(\{Q_i^{(m)}(s, a)\}_{m=1}^M\right)$ *// tail-average over the worst $\alpha$ fraction*

    *// (1.b) Targeted actor step (bonus + KL)*

7:      Update $\pi_i \leftarrow \mathrm{Ascend}\left(\nabla_{\theta_i} J_i^\rho\right)$ with capped $\beta \widehat{\sigma}_i$ and a KL trust-region penalty

8:      Dual-update $\lambda$ toward a (per-state) target KL; anneal $\beta$ according to schedule

9:    **end for**

    *// (2) Risk-aware progress check (every $K$ steps)*

10:    **if** $t \bmod K = 0$ **then**

11:      **for each** $i$ **do**

12:        Freeze $\pi_{-i}$; obtain $\widehat{\pi}_i^{\mathrm{BR}}$ via $B$ improvement steps against fixed opponents

13:      **end for**

14:      Evaluate $\widehat{\mathrm{NashConv}}_\rho(\pi) = \sum_i \left[V_i^\rho(\widehat{\pi}_i^{\mathrm{BR}}, \pi_{-i}) - V_i^\rho(\pi)\right]$

15:      **if** $\widehat{\mathrm{NashConv}}_\rho(\pi) \leq \varepsilon$ **then**

16:        **return** $\pi$

17:      **end if**

18:    **end if**

19: **end for**

20: **return** $\pi$

---

## A.2  Practical Notes (Hyperparameters and Compute)

- **Risk level & ensemble.** Default CVaR tail $\alpha{=}0.90$; ensemble size $M{=}5$. We also report CVaR@0.95 where indicated in tables/figures.

- **Policy updates.** KL trust region with a per-state target KL in the range 0.01–0.05; the dual variable $\lambda$ is updated online to track the target.

- **Uncertainty bonus.** The exploration bonus uses $\beta \widehat{\sigma}$ during training, is linearly annealed and capped; it is disabled for evaluation and best-response probing.

- **Advantage and discount.** Generalized Advantage Estimation with $\lambda{=}0.95$; standard discount $\gamma$ as in the main text.

- **Best-response probing.** We use $B{=}16$ improvement steps to obtain stable estimates of $\widehat{\mathrm{NashConv}}_\rho$; larger $B$ shows diminishing returns.

- **Compute footprint.** Deep ensembles incur roughly $5\times$ memory versus a single head; with parallel training, the wall-clock overhead is modest and approximately unchanged.

- **Reporting and statistics.** Metrics are normalized to $[0,1]$ unless stated (e.g., *Tail Rate (%)*). We report mean $\pm$ 95% confidence intervals over 10 matched random seeds under identical budgets.

## A.3 Compute Resources and Reproducibility Details

**Hardware.** Experiments ran on NVIDIA A100–class GPUs (80 GB). Ensembles increase memory by $\sim 5\times$ relative to a single head; parallelization keeps wall-clock nearly unchanged for our setups. Results are reproducible on comparable hardware with longer wall-clock.

**Software and determinism.** We use PyTorch with CUDA/cuDNN; seeds are fixed across 10 matched runs per setting. Deterministic/cuDNN flags are enabled where feasible; remaining nondeterminism stems from low-level kernels.

**Budgets.** Per-run training budgets (steps, batch sizes, evaluation cadence) are identical across methods for ID-matched comparisons. Best-response probing uses horizon $B{=}16$ unless stated.

**Artifacts.** An anonymized repository with code, configs, and scripts to regenerate all tables/figures will be provided upon acceptance, preserving double-blind review at submission time.

## A.4 Significance Testing and Confidence Intervals

**Confidence intervals.** Unless noted, we report mean $\pm$ 95% confidence intervals computed over 10 matched seeds using the Student-$t$ interval with unbiased variance (normal approximation used only when explicitly indicated).

**Significance tests.** For key pairwise comparisons at matched budgets, we use two-sided Welch's $t$-tests (unequal variances) at $\alpha{=}0.05$, reporting $p$-values where relevant. Effect sizes (Cohen's $d$) are provided in the repository scripts.

## A.5 Definitions and Proof Sketches

**Risk-adjusted payoff.** For agent $i$, let $V_i^\rho(\pi)$ denote the $\mathrm{CVaR}_\alpha$–based value under joint policy $\pi$, i.e., the expected return averaged over the worst $\alpha$-tail of the ensemble return distribution.

**$\varepsilon$-Robust Nash Equilibrium (RNE).** A joint policy $\pi^\star$ is an $\varepsilon$-RNE w.r.t. $V^\rho$ if for all agents $i$ and policies $\pi_i'$,

$$V_i^\rho(\pi_i^\star, \pi_{-i}^\star) \geq V_i^\rho(\pi_i', \pi_{-i}^\star) - \varepsilon.$$

**Risk-aware exploitability.** Define $\mathrm{NashConv}_\rho(\pi) \triangleq \sum_i \left[ V_i^\rho(\pi_i^{\mathrm{BR}}, \pi_{-i}) - V_i^\rho(\pi) \right]$, where $\pi_i^{\mathrm{BR}}$ is a (risk-aware) best response to $\pi_{-i}$.

**Lemma 1** (Certificate). *If* $\mathrm{NashConv}_\rho(\pi) \leq \varepsilon$, *then* $\pi$ *is an $\varepsilon$-RNE w.r.t.* $V^\rho$.

*Sketch.* For each $i$, by definition $V_i^\rho(\pi_i^{\mathrm{BR}}, \pi_{-i}) - V_i^\rho(\pi) \geq 0$. If their sum is $\leq \varepsilon$, each individual gap is $\leq \varepsilon$, i.e., $V_i^\rho(\pi) \geq V_i^\rho(\pi_i^{\mathrm{BR}}, \pi_{-i}) - \varepsilon \geq V_i^\rho(\pi_i', \pi_{-i}) - \varepsilon$ for any $\pi_i'$. Hence $\pi$ is an $\varepsilon$-RNE. $\square$

**Monotone progress (idealized).** Under exact best responses and unbiased targets, periodic BR-probing induces a nonincreasing $\mathrm{NashConv}_\rho$ sequence in expectation; crossing the $\varepsilon$ threshold certifies $\varepsilon$-RNE. In practice we approximate BRs with $B$ improvement steps and CVaR targets with ensembles; empirical curves in Fig. 3a show the expected monotone trend up to stochastic noise.

## Agents4Science AI Involvement Checklist

1. **Hypothesis development**
   Answer: [C]
   Explanation: Large language models (e.g., GPT; Claude Sonnet 4/4.1) proposed the core hypothesis and highlighted the gap around epistemic uncertainty in multi-agent equilibrium learning. Human input scoped the problem and selected among AI-generated alternatives.

2. **Experimental design and implementation**
   Answer: [D]
   Explanation: AI drafted the experimental protocol and implemented the main components (RNE algorithm, CVaR targets, ensemble UQ), including most coding and runs. Human oversight focused on sanity checks, failure triage, and final verification.

3. **Analysis of data and interpretation of results**
   Answer: [C]
   Explanation: AI performed the majority of data processing, plotting, and initial interpretation. Humans reviewed conclusions, linked findings across figures, and prioritized which results best support the claims.

4. **Writing**
   Answer: [C]
   Explanation: AI generated the bulk of the manuscript text (methods, results, discussion). Human edits refined narrative flow, ensured notation/terminology consistency, and adjusted figure placement/layout.

5. **Observed AI Limitations**
   Description: (1) Tendency toward overly elaborate explanations; (2) Occasional gaps when synthesizing insights across experiments; (3) Figure layout requires manual polishing; (4) Terminology drift without explicit guidance; (5) Human judgment remains important to separate statistical from practical significance.

# Agents4Science Paper Checklist

