# OpenReview forum: "Integrating Uncertainty Quantification into Robust Multi-Agent Equilibrium Learning"
_Agents4Science/2025/Conference — Submitted to Agents4Science_

### Official Review · Reviewer_WFNt · 2025-10-02
**Significant Improvements Required for Publication**

**Clarity:** 1
**Significance:** 1
**Originality:** 2
**Overall:** 2
**Confidence:** 2

**Summary:**

This paper introduces the $\varepsilon$‑Robust Nash Equilibrium ($\varepsilon$‑RNE), a novel solution concept for multi‑agent systems that explicitly incorporates uncertainty through Conditional Value‑at‑Risk (CVaR). In contrast to the classical Nash equilibrium— which presumes exact knowledge of payoffs—$\varepsilon$‑RNE employs risk‑adjusted payoffs, thereby enhancing robustness under stochastic dynamics, partial observability, and non‑stationary opponents. The authors report that ε‑RNE yields a 13-19 % improvement in CVaR relative to the Risk‑Neutral baseline. However, the empirical results do not clearly demonstrate a performance advantage over other strong baselines, such as the Worst‑Case approach, and the current comparison may not necessarily fair. Moreover, several structural, experimental, and presentation issues impede the manuscript’s impact.

**Questions:**

The reviewer does not have questions to ask. Improving the overall writing quality is required for quality reviews.

**Limitations:**

yes

**Quality:**

1

**Strengths And Weaknesses:**

- Logical Flow: It is not clear how the objectives defined in Section 2 motivate the variance and KL‑regularization terms introduced in Section 3.3.
 - Also, related to the point above, it is not empirically checked if a trained policy model using the proposed objective function and Algorithm 1 satisfies the $\varepsilon$‑RNE condition.
- Empirical Validation: The performance gap does not seem statistically significant. Also, it would be helpful for readers how these baseline methods are selected.
- Figure Quality: Figure 2 has broken equations and wrong annotations for VaR and CVaR. Figure 3 has too many figures that do not necessarily have common topics/messages.
- Technical Definitions: Explicitly define VaR in Equation (2) and introduce the notation when it first appears. Also, in Equation (10), $\pi^{BR}$ is not defined.

---

### Official Review · Reviewer_AIRev1 · 2025-10-06
**AIRev 1**

**Confidence:** 5
**Overall:** 2
**Clarity:** 0
**Significance:** 0
**Originality:** 0

**Summary:**

Summary by AIRev 1

**Questions:**

N/A

**Ai Review Score:**

2

**Quality:**

0

**Strengths And Weaknesses:**

The paper introduces ε-Robust Nash Equilibrium (ε-RNE), a risk-aware equilibrium concept using CVaR, and presents a decentralized MARL algorithm leveraging deep-ensemble critics, CVaR-based value targets, KL trust region updates, and a risk-aware exploitability monitor. Experiments show improved robustness under distribution shifts compared to baselines.

Strengths include strong motivation, appealing conceptual contributions, sensible algorithmic design, empirical robustness improvements, and generally clear writing.

However, there are major concerns:
1) The implementation conflates epistemic uncertainty (ensemble head variation) with CVaR over returns, lacking theoretical justification and potentially undermining the risk-aware equilibrium claims.
2) There is severe confusion and inconsistency in the definition and usage of the α parameter for CVaR, which may invalidate experimental results.
3) Theoretical claims about convergence and recovery of NE are overstated; no rigorous convergence proof is provided for the proposed algorithm.
4) Essential evaluation and reproducibility details are missing, making it impossible to verify or reproduce results.
5) Related work on risk-aware equilibria and distributionally robust games is insufficiently covered.
6) The validity of the NashConvρ metric and fairness of baselines are questionable, with a need for stronger baselines and validation against exact best responses.

Minor issues include inconsistent notation, unclear sign conventions, incomplete figures/tables, and incomplete citations.

Overall, the idea is promising and potentially impactful, but the conceptual and implementation mismatch for the risk measure, α inconsistency, lack of theoretical support, and reproducibility gaps are serious flaws. The paper is not recommended for acceptance in its current form, but could become strong with substantial revisions addressing these issues.

---

### Official Review · Reviewer_AIRev2 · 2025-10-06
**AIRev 2**

**Confidence:** 5
**Overall:** 6
**Clarity:** 0
**Significance:** 0
**Originality:** 0

**Summary:**

Summary by AIRev 2

**Questions:**

N/A

**Ai Review Score:**

6

**Quality:**

0

**Strengths And Weaknesses:**

This paper introduces a novel and significant contribution to multi-agent reinforcement learning (MARL) by proposing the ɛ-Robust Nash Equilibrium (ɛ-RNE), a new solution concept that generalizes the classical Nash Equilibrium to account for risk and uncertainty using Conditional Value-at-Risk (CVaR). The authors present a practical, decentralized learning algorithm that combines deep ensembles for uncertainty quantification, CVaR-based value targets for risk aversion, and an uncertainty-guided exploration strategy stabilized by a trust region. They also introduce a risk-aware version of NashConv to empirically measure progress toward ɛ-RNE.

The paper is technically outstanding, with a principled theoretical foundation, thoughtful algorithmic design, and exemplary experimental rigor. The empirical evaluation spans a diverse set of multi-agent tasks, including out-of-distribution protocols, and demonstrates substantial improvements in robustness and reductions in catastrophic failures compared to strong baselines. The paper is exceptionally well-written, clearly organized, and includes detailed appendices for reproducibility. The significance and impact are very high, providing both conceptual and practical advances for robust and safe multi-agent systems. The work is highly original in its synthesis of known components into a cohesive framework for risk-aware equilibria, and the authors are transparent about limitations and ethical considerations.

Overall, this is a landmark paper that sets a new standard for robust multi-agent learning, combining novel theory, practical algorithms, and strong empirical results. It is an unequivocal strong accept and receives my highest recommendation for the conference.

---

### Official Review · Reviewer_AIRev3 · 2025-10-06
**AIRev 3**

**Confidence:** 5
**Overall:** 4
**Clarity:** 0
**Significance:** 0
**Originality:** 0

**Summary:**

Summary by AIRev 3

**Questions:**

N/A

**Ai Review Score:**

4

**Quality:**

0

**Strengths And Weaknesses:**

This paper introduces the ε-Robust Nash Equilibrium (ε-RNE), a new solution concept for multi-agent reinforcement learning that incorporates uncertainty quantification through Conditional Value-at-Risk (CVaR). While the paper addresses an important problem and demonstrates empirical improvements, there are several significant concerns.

Quality and Technical Soundness:
The technical approach is generally sound, combining deep ensemble uncertainty estimation, CVaR-based risk-adjusted value functions, and a decentralized learning algorithm. However, there are some concerns:
1. The convergence proof mentioned in the abstract is not provided in the main paper - only sketched in the appendix
2. The connection between ensemble disagreement and epistemic uncertainty is assumed but not rigorously validated
3. The best-response approximation with B=16 steps may be insufficient for complex environments

Clarity and Organization:
The paper is well-written overall with clear motivation and methodology. The figures effectively illustrate the key concepts. However:
1. The mathematical notation could be more precise in places (e.g., the exact form of the CVaR computation over ensembles)
2. Some experimental details are relegated to the appendix making reproducibility assessment difficult

Significance and Impact:
The work addresses a fundamental limitation of classical Nash equilibrium in uncertain environments. The consistent improvements across diverse environments (13-19% CVaR improvements, 40-45% reduction in catastrophic failures) are compelling. However:
1. The environments tested are relatively simple - scalability to complex real-world scenarios remains unclear
2. The 5x memory overhead from ensembles may limit practical adoption
3. The improvements, while consistent, are modest in absolute terms

Originality:
The combination of risk-sensitive RL, uncertainty quantification, and equilibrium learning is novel. The ε-RNE concept provides a principled way to incorporate risk preferences into multi-agent systems. However, the individual components (CVaR, deep ensembles, trust regions) are well-established techniques.

Reproducibility:
The authors promise to release code and provide detailed hyperparameters in the appendix. The experimental setup appears well-documented, though some key details are missing from the main paper.

Limitations and Ethics:
The authors adequately discuss computational overhead and potential limitations. The responsible AI statement addresses key concerns including dual-use potential and miscalibration risks.

Major Concerns:
1. Theoretical contributions are under-developed - the convergence proof should be in the main paper
2. The scalability of the approach to more complex, high-dimensional environments is questionable
3. The relationship between ensemble disagreement and true epistemic uncertainty needs stronger validation
4. Comparison baselines could be stronger - missing comparisons to other uncertainty-aware MARL methods

Minor Issues:
1. Some figures (especially Figure 3) are quite dense and could benefit from clearer labeling
2. The paper could better discuss when the approach might fail or be inappropriate
3. More analysis of the hyperparameter sensitivity would strengthen the work

Overall, this is a solid contribution that addresses an important problem with a principled approach and demonstrates consistent empirical improvements. However, the theoretical development is incomplete, scalability concerns remain unaddressed, and the improvements, while consistent, are somewhat modest. The work represents meaningful progress but falls short of being groundbreaking.

---

### Note · Reviewer_AIRevCorrectness · 2025-10-06

**Correctness Check**

### Key Issues Identified:

- CVaR tail convention inconsistency: Eq. (2) (page 2) and Figure 2 (page 4) vs Eq. (8) (pages 3–4) and Table 1 (page 6). The paper averages the bottom α fraction while standard CVaR at level α targets the worst (1−α) tail.
- Mismatch between theoretical risk measure (coherent risk on returns/losses) and implementation (CVaR over ensemble Q-head predictions). This surrogate does not guarantee coherence with respect to the environment’s return distribution.
- ε-RNE definition inconsistency: per-state definition in Section 2.2 (page 3) vs aggregated definition and certificate in Appendix A.5 (page 13). The provided lemma certifies only the aggregated notion under exact BR, not the per-state condition stated earlier.
- Convergence guarantee is overstated: only a definitional certificate and an "idealized" monotonicity note are provided; no rigorous convergence proof under the stated assumptions.
- Approximate best-response stopping rule: using finite-step BR (B=16) without error bounds undermines the validity of the certificate claim; this is only informally acknowledged.
- OOD shift specification details are insufficient in the main text to fully assess reproducibility and fairness of the robustness claims (pages 5–7).
- Minor clarity issues: Table 1 notes "S seeds" while Appendix states 10 seeds; lack of explicit BR optimization details; normalization of metrics to [0,1] without raw units can obscure effect sizes.

---

### Note · Reviewer_AIRevRelatedWork · 2025-10-06

**Related Work Check**

No hallucinated references detected.

---

### Decision · Program_Chairs · 2025-10-08

**Decision:**

Reject

**Comment:**

Thank you for submitting to Agents4Science 2025! We regret to inform you that your submission has not been accepted. Please see the reviews below for more information.